# Assessing Influence Mechanism of Management Overconfidence, Corporate Environmental Responsibility and Corporate Value: The Moderating Effect of Government Environmental Governance and Media Attention

**DOI:** 10.3390/ijerph20010577

**Published:** 2022-12-29

**Authors:** Guiyu Bai, Delin Meng

**Affiliations:** 1Business School, University of Jinan, Jinan 250002, China; 2School of Economics and Management, Dalian University of Technology, Dalian 116024, China

**Keywords:** management overconfidence, corporate environmental responsibility, environmental governance, media attention, corporate governance

## Abstract

China’s economic development has gradually entered a new period of slowing down and changing from quantity to quality, which has put forward higher requirements for environmental quality. How to better fulfill environmental responsibilities and realize a virtuous circle of “environmental protection for development” and a value growth model are essential issues that enterprises should consider and solve. Overconfidence, as one of the significant psychological characteristics of management, has caused more and more attention to its economic consequences. In order to clarify the internal logical relationship between the two and help enterprises optimize their environmental responsibility decisions, the paper is based on upper echelon theory and stakeholder theory. It focuses on the micro-situation of the corporate, empirically testing the influence of management overconfidence on corporate environmental responsibility by using the OLS regression analysis method, taking the manufacturing listed companies in the Shanghai and Shenzhen Stock Exchange of China from 2010 to 2017 as the research sample. The study discusses the moderating effect of government environmental governance and media attention on the relationship between management overconfidence and corporate environmental responsibility. The empirical results show a negative correlation between management overconfidence and corporate environmental responsibility. Both government environmental governance and media attention will weaken the negative correlation between management overconfidence and corporate environmental responsibility. Further research finds that management overconfidence has a weakening effect on corporate value, and corporate environmental responsibility plays a partial mediating role between management overconfidence and corporate value.

## 1. Introduction

Environmental degradation will become one of the greatest challenges facing human development in the future [1,2]. China’s economy is changing from a high-speed to a high-quality development stage. The extensive and epitaxial development model of “pollution and treatment at the same time” has resulted in a waste of resources and high environmental pollution, which has seriously hindered the transformation and upgrading of the economic structure. Therefore, implementing “lucid waters and lush mountains are invaluable assets” and constructing “beautiful China” has become the unavoidable focus in China’s transformation of development mode and optimization of economic structure. From the general layout of the construction of ecological civilization in the 19th National Congress Report of China to “to continuously improve the ecological environment and realize new progress in the construction of ecological civilization” mentioned in the 14th Five-Year Development Plan, the various measures in the top-level design have demonstrated the increasing concern of the Chinese government on environmental issues. In addition, as the primary pollutant discharge and the critical link of the national environmental protection blueprint, how should enterprises respond to the national environmental protection call while pursuing the maximization of operating profits, incorporate environmental protection into their own strategic decisions, fulfill their environmental responsibilities, win the environmental defense war with the country through practical actions, and realize the virtuous circle of “promoting development with environmental protection” has become an essential topic of common concern in academic and practical circles [3].

Corporate environmental responsibility refers to the behavior that enterprises adopt to meet the requirements of environmental ethics and laws in order to realize the sustainable development of the environment [4], which to some extent reflects the degree to which enterprises actively respond to the requirements of stakeholders and actively participate in social initiatives [5,6,7]. With the deepening of the construction of ecological civilization, stakeholders have gradually put forward more urgent and specific requirements for enterprises’ environmental responsibilities and transfer such requirements to enterprises in the form of legitimacy pressure. Enterprises are required to increase investment in environmental protection and effectively participate in environmental management to respond [8,9,10]. In recent years, many pieces of research have emerged to explore the influence of management’s characteristics on corporate decision-making. The upper echelon theory is the core theory to explain the influence of management’s attributes on corporate results, which has been widely used in management [11,12]. The upper echelon theory holds that senior managers, as the main body of corporate strategic decisions, will make highly personalized interpretations according to their own organizational situations and adopt strategic behaviors with individual characteristics according to their own cognition, values and experience, in other words, corporate decisions will have obvious “manager effect” [13]. However, there is very little literature to study the attitude and influence on CER from the perspective of management’s personality traits. Many studies have found that management’s personality traits will significantly affect the business decisions of enterprises [14,15,16]. Therefore, it is necessary to explore the behavioral motivation and impact mechanism of management characteristics on corporate environmental responsibility from the micro-level of the enterprise. Previous studies have shown that the overconfidence of the management, as an important embodiment of the characteristics of the management in the psychological level, will have a direct impact on strategic decisions such as investment, M&A [17,18,19,20], and technological innovation [21,22,23,24]. It is a fascinating topic to explore the influence of management’s overconfidence on CER because when management has overconfidence, this persistent character may affect the resources they own and their attitudes towards external stakeholders, which in turn affects the environmental decision-making of the enterprise.

Our research objective is to explore the impact of executive overconfidence on CER in listed companies in emerging market countries and the contingent impact of the two effects based on the upper echelon theory and stakeholder theory. Specifically, by collecting data from listed companies in China, we use OLS regression and other research methods to explore the relationship between management overconfidence and CER. Considering the legitimacy pressure of external stakeholders, we also introduce two variables, government environmental governance and media attention, to explore the influence boundary of environmental decision-making under management overconfidence. Compared with the existing literature, the possible research contributions and research innovations are as follows: Firstly, the research explores the psychological characteristics of the management—overconfidence’s attitude towards the enterprise’s environmental decision-making, which not only can expand the application paradigm of the upper echelon theory and enrich the management decision-making research literature in the field of management’s personality characteristics, but also explores the motivation of corporate environmental responsibility decision-making in corporate governance and deepens the understanding of the influencing factors of CER at the micro level. Secondly, based on the stakeholder theory, we consider the legitimacy problems faced by enterprises. Given the institutional situation in China, we try to distinguish the external stakeholders in variable selection and measurement from the government’s perspective and external media attention. By exploring the requirements of different stakeholders for continuous improvement of the environment and the differential impact mechanism of external legitimacy pressure on corporate strategic decisions, the research clarifies the role path of external stakeholders in corporate governance and deepen and innovate the logic of current literature research on the impact of external stakeholders on corporate decision-making. Finally, the research conclusion not only expands the upper echelon theory and stakeholder theory on the theoretical level but also provides new ideas and empirical evidence for developing countries with similar institutional and cultural patterns on how to view the relationship between executive personality and corporate decision-making.

## 2. Theoretical Analysis and Hypothesis Development

### 2.1. Influence Mechanism of Corporate Environmental Responsibility

Management overconfidence is a representative corporate governance factor can affect corporate environmental responsibility in the following ways:

First of all, overconfident management will overestimate the resource endowments of enterprises and firmly believe that they have more hidden information than they actually have, and even more than other competitors [14,22,23]. Based on this recognition, the management will underestimate the dependence on the resources of external stakeholders, believing that the enterprise has a good resource reserve and strong development potential, thus reducing the attention to the environmental protection needs of external stakeholders and unwilling to meet external requirements and fulfill the corresponding corporate environmental responsibility [24,25].

Secondly, due to the existence of a specific self-attribution bias [26], overconfident management pays more attention to individual success and performance returns when making decisions [27], is more inclined to make challenging decisions and judgments driven by a high degree of self-identification, is keen on higher-difficulty projects and highly complex transactions, obtains lasting satisfaction through completing challenging tasks, and proves that it does have “above-average” management capabilities [28]. Corporate environmental responsibility is different from potentially high-yield risky strategic decisions in terms of implementation path and benefit return. The rigid characteristics such as a long revenue cycle and uncertain benefit performance make it difficult for management to obtain satisfaction of individual performance returns and weaken management’s sense of self-efficacy and identity. As a result, overconfident management is reluctant to devote too much effort to fulfilling corporate environmental responsibility.

Finally, the stakeholder theory holds that there is a contractual relationship between enterprises and stakeholders, and the existence of such a contract is the basis for the interaction between enterprises and stakeholders [29]. Under the contractual framework of external stakeholders, enterprises need to strengthen the interaction with the external environment, carry out corresponding environmental management, and accept the suggestions and supervision of external stakeholders. However, overconfident management is often self-centered, believing that fate is in their own hands, obsessed with self-set results, and unwilling to be interfered and constrained by external factors [30,31,32,33]. For the overconfident management, the intervention of the voice of external stakeholders will bring certain constraints to the management’s self-set planning and behavior in the cognitive model and path, affecting the management’s judgment and decision on the development layout of the enterprise, thus causing the management to have certain psychological exclusion to the stakeholders, which will also bring a negative impact on the corporate environmental responsibility. Therefore, based on the above discussion, we propose the first study hypothesis as follows:

**Hypothesis 1 (H1).** *Management overconfidence is negatively correlated with corporate environmental responsibility*.

### 2.2. The Moderating Role of Government Environmental Governance

The legitimacy theory indicates that enterprises should engage in reasonable and legitimate activities under the norms, values and belief systems constructed by society [34]. In other words, in order to obtain legitimacy, enterprises must obtain the understanding and recognition of key stakeholders in the market competition, conform to the values, behavior norms and expectations of key stakeholders, and meet their interests [35,36,37]. China’s economy has shifted from high-speed to high-quality growth, and the requirements for ecological sustainable development and environmental improvement are increasingly high, the government also increasingly calls for enterprises to fulfill their corresponding responsibilities in environmental protection [1], which has brought higher legitimacy pressure to enterprises. In 2020, China issued the “Guiding Opinions on the Construction of a Modern Environmental Governance System”, which pointed out that the construction of a modern environmental governance system should focus on enterprises, perfect the responsibility system of enterprises in environmental governance, promote the greening of enterprise production services, and improve the ability and level of pollution control. Therefore, under the pressure of the legitimacy of the government, enterprises must abide by the corresponding environmental management system, conduct green operation, promote green development and achieve a “win-win” of economic and environmental benefits.

We hold the view that the overconfident management will weaken the psychological exclusion of enterprises to fulfill their environmental responsibilities under the situation of strong government environmental governance. The reasons are as follows: Firstly, as one of the core stakeholders of an enterprise, the government can have an important impact on the survival and development of the enterprise through policies and regulations, resource allocation, macro-control and other means [38]. If corporate decision-making deviates from government rules, it will be difficult to obtain government support in terms of capital subsidies, tax incentives and credit support [39,40], which poses a certain threat to the development and even survival of the enterprise. Secondly, Hofstede’s theory of cultural dimensions points out that China’s culture shows a greater power distance and advocates collectivism [41]. Under this cultural background, senior executives will pay more attention to the information conveyed by the state and society and think that the behavior and attitude of external stakeholders will be closely related to their own decisions [42], and they are also easier to accept the suggestions of external stakeholders. Therefore, the stronger the government’s environmental governance level is, the overconfident management will pay more attention to the value and role of external stakeholders, re-examine its own resource endowments, internal and external opportunities and threats, cater to the government’s requirements for environmental governance, choose to solicit more stakeholders’ support, and weaken the previous negative attitude towards corporate environmental responsibility. Based on the above discussion, we defined the second hypothesis as:

**Hypothesis 2 (H2).** *Government environmental governance weakens the negative correlation between management overconfidence and corporate environmental responsibility*.

### 2.3. The Moderating Effect of the Media Attention

As an information transmission medium for stakeholders, the media can influence corporate behavior through two paths. On the one hand, as one of the three major public pressures [43], the media’s negative reports on corporate behavior will play a role of deterrence and supervision, which will greatly stimulate corporate nerves and bring heavy pressure on corporate legitimacy. Under the pressure of negative media reports, enterprises will try their best to perform more pro-social behaviors to gain the understanding and recognition of stakeholders, reduce the external risks of enterprises and maintain a positive image. On the other hand, as the public likes companies with responsibility awareness and hopes that these companies can become more and more successful [23], the media will also be more inclined to report positively on the positive behavior of the companies [44], helps enterprises to have a two-way dialogue with the public, so that enterprises can better absorb public opinions and suggestions and eliminate public prejudice, thus improving the legitimacy of enterprises, gaining more recognition and favor from stakeholders, capturing more tangible and intangible resources [45].

First of all, when media attention is high, overconfident management will bear more legitimacy pressure and have to face feedback from external situations to adjust their own decisions [23,46,47], thinking about the consequences of corporate behavior is more comprehensive and cautious, and can realize the value return brought by the fulfillment of environmental responsibilities more objectively and soberly. Secondly, the tendency of the management to rely on their own management ability to improve the value of the enterprise will be further amplified due to more media attention, and the intention to prove that they are above average is also stronger. Because corporate environmental responsibility is desirable for stakeholders such as the public and investors [48], and stakeholders also have a special attribution effect, which tends to attribute the behavior and results of the enterprise to the managers rather than the owners [49,50], management will be more likely to benefit from positive corporate events. Therefore, the overconfident management may choose to fulfill their corporate environmental responsibilities to improve their exposure and reputation under more media attention to prove their management ability in this special way.

Based on the impact path of the above two media concerns, we believe that the overconfident management will re-examine the positive externalities brought by the corporate environmental responsibility, instead of only considering its rigid characteristics, thus weakening the negative intention for the corporate environmental responsibility. Based on this, the third hypothesis is put forward here:

**Hypothesis 3 (H3).** *Media attention weakens the negative correlation between management overconfidence and corporate environmental responsibility*.

The study model is shown in Figure 1.

## 3. Materials and Methods

### 3.1. Data and Sample Selection

The research selects manufacturing listed companies (industry code: C13–C42) in Shanghai and Shenzhen stock exchanges from 2010 to 2017 as the research sample. These listed companies are distributed in Guangdong, Zhejiang, Jiangsu, Beijing, Shanghai, Shandong and other provinces and cities in China. Then, we conducted the following processing: (1) no significant restructuring phenomenon such as the change of controlling shareholder occurred during the sample observation period; (2) excluding financial sample companies; (3) excluding ST or * ST companies; (4) eliminating companies with severe data deficiency; (5) to avoid the influence of outliers, the above continuous variables were winsorized. The data of CER are from Hexun CSR report evaluation index, the data of government environmental governance and media attention were collected manually by the authors. All these data were officially released by the enterprise, and their authenticity and reliability are trustworthy. The other data were collected from the CSMAR database. In this paper, empirical analysis method was used to verify the hypothesis. With reference to previous research, variables were collected and sorted out, and then a statistical model corresponding to the hypothesis was constructed. Then, we chose the STATA16.0 measurement software to perform regression analysis on the selected data. The STATA software has full-featured panel data processing capabilities, and the results show that it is more intuitive, which has been widely used in relevant empirical research.

### 3.2. Variable Definitions

#### 3.2.1. Dependent Variable

In this paper, corporate environmental responsibility is taken as a dependent variable. Drawing on the practice of Zeng et al. [51], based on the accuracy and availability of data, the third-party scoring index is selected according to the environmental activities disclosed in each listed company’s annual report. The environmental responsibility score in Hexun CSR score was used to measure corporate environmental responsibility. Hexun corporate environmental responsibility scores are derived from the CSR reports and annual reports published by the Shanghai and Shenzhen stock exchanges and are scored according to five aspects: environmental awareness, environmental management system certification, environmental investment, pollutant emission types and energy-saving types. The scoring system provides China with the results of environmental responsibility scores for all listed companies. The research samples are comprehensive and complete and can effectively avoid sample selectivity bias.

#### 3.2.2. Independent Variable

The independent variable in this paper is management overconfidence (OC). This paper refers to the research of Brown and Sarma [52], Hayward and Hambrick [53], considering the actual situation of China’s capital market and the availability of data, selects managers’ relative compensation to measure management overconfidence. The specific measurement method involved ranking the management’s remuneration from high to low, calculating the ratio of the top three management’s remuneration to the sum of all remuneration, and treating it as a dummy variable according to the median. If the value obtained is higher than the median, the management is deemed to be overconfident, and the value is 1; otherwise, it is 0.

#### 3.2.3. Moderating Variables

In this paper, government environmental governance (EG) is regarded as the first regulatory variable. There is no unified measurement standard for environmental governance in the existing literature. Scholars mainly measure it through indicators such as unit output pollution emissions and pollution investment costs. Considering that environmental governance has various manifestations such as administrative orders and mandatory constraints, it is difficult to effectively measure government policies in an all-round way only by a single indicator. Therefore, this paper draws on the method of Chen et al. [54]: manually collect and sort out the word frequency of the relevant words with the word environmental protection in the work report of each provincial government in China, divide the word frequency with the full text of the work report of the provincial government, and then match the variable with the province where the enterprise is located as an agent variable of environmental governance. Among them, the vocabulary related to “environmental protection” includes 93 words such as environment, pollution, environmental protection, environmental regulation, environmental governance, environmental quality, environmental pollution, and energy conservation and emission reduction. The advantage of selecting this variable is that, on the one hand, it can more comprehensively reflect the degree and overall picture of the government’s environmental governance. On the other hand, because the government work report is generally released at the beginning of the year, and economic activities run through the year, it can more effectively alleviate endogenous problems.

Media attention (Media) is regarded as the second regulatory variable. Most research selects newspapers, journals and other paper media to measure media attention. However, with the rapid development of the Internet, online media has gradually become the main way for the public to understand the information of listed companies. Therefore, this paper uses the advanced search function of Baidu News, substitutes the short names of securities of each listed company one by one and performs a time-interval search to obtain the total number of news reports of each year, and then adds 1 to take natural logarithm as a measure of media attention.

#### 3.2.4. Control Variables

Based on previous studies [23,25], adopt the following control variables, namely, management shareholding (MS), board independence (Independence), capital structure (Lev), the company’s performance (ROA), the board leadership structure (Duality), equity concentration (EC), age of the company (Age), growth of the company (Growth) and year. Table 1 shows the definition and measurement methods of variables in this paper. Table 1 summarizes the full list of variables used in this study.

### 3.3. Empirical Model Design

In order to test the hypothesis in this paper, the following multiple regression model is designed.
CER_i,t_ = α_0_ + Σα_q_Control_i,t_ + ε(1)
CER_i,t_ = α_0_ + α_1_OC_i,t_ + Σα_q_Control_i,t_ + ε(2)
CER_i,t_ = α_0_ + α_1_OC_i,t_ + α_2_EG_i,t_ + α_3_OC × EG_i,t_ + Σα_q_Control_i,t_ + ε(3)
CER_i,t_ = α_0_ + α_1_OC_i,t_ + α_2_Media_i,t_ + α_3_OC × Media_i,t_ + Σα_q_Control_i,t_ + ε(4)

Among them, model 1 is the regression of control variables to corporate environmental responsibility; model 2 is the regression of management overconfidence to corporate environmental responsibility, which can be used to test hypothesis 1; model 3 adds the interaction between government environmental governance and management overconfidence and government environmental governance on the basis of model 2, which is used to analyze the moderating effect of government environmental governance on the relationship between management overconfidence and corporate environmental responsibility, and can verify hypothesis 2; model 4 adds media attention and the interaction between management’s overconfidence and media attention on the basis of model 2, which is used to analyze the moderating effect of media attention on the relationship between management’s overconfidence and corporate environmental responsibility, which can verify hypothesis 3.

## 4. Results

### 4.1. Descriptive Statistics

As can be seen from Table 2, the minimum value of the CER (corporate environmental responsibility) score is 0, the maximum value is 25, and the average value is only 2.047, indicating that there is a big gap in the performance of the environmental responsibility of manufacturing enterprises. Many enterprise managements lack a high awareness of environmental protection and do not perform their environmental responsibilities well. The average value of management overconfidence is 0.488. It can be seen that the psychological characteristic of management overconfidence is more common at the enterprise level. The average value of government environmental governance is 0.015, the maximum value is 0.026, and the standard deviation is small, which indicates that each province has a certain degree of attention to environmental protection, and the difference between provinces is small. The minimum value of media attention is 0, the maximum value is 12.660, and the average value is 2.927, indicating that different manufacturing enterprises have certain differences in media attention.

At the end of this section, we report Spearman correlation coefficient and variance expansion coefficient. The results are shown in the following table. As can be seen from Table 3, the correlation coefficient between management overconfidence and corporate environmental responsibility is negative, which preliminarily proves our hypothesis that management overconfidence can reduce the degree of corporate environmental responsibility fulfillment. In addition, the correlation coefficients in Table 3 are all lower than 0.45, and the average value of the variance expansion coefficients shown in Table 4 is 1.14, which indicates that the variables used herein are not affected by multicollinearity.

### 4.2. Multiple Regression Results

According to the model designed above, the regression analysis is performed on the data by using Stata15.0 software. The specific calculation results are shown in Table 5. Model 1 lists the regression results of the control variables on the corporate environmental responsibility. Model 2 is the regression analysis of the managerial overconfidence variable on the corporate environmental responsibility. It can be seen that the regression coefficient between managerial overconfidence and corporate environmental responsibility is −0.904, and is significantly negative on the basis of 1%, which proves Hypothesis 1. Model 3 is a regression analysis of the moderating effect of government environmental governance on the relationship between management’s overconfidence and corporate environmental responsibility. The results show that the interaction term between management’s overconfidence and government environmental governance is 0.236, which is significantly positive on the basis of 5%, indicating that Hypothesis 2 has been verified. Model 4 is a regression analysis of the moderating effect of media attention on the relationship between management overconfidence and corporate environmental responsibility. The results show that the interaction term between management overconfidence and media attention is 0.156, which is significantly positive on the basis of 10%. Therefore, Hypothesis 3 is also verified.

### 4.3. Robustness

#### 4.3.1. Random Sample

Considering that the number of samples may interfere with the significance of the conclusion, according to the research of Li et al. [55], this paper randomly selects 80% of the samples to test. The test results are shown in Table A1 in model 2, the regression coefficient between management overconfidence and corporate environmental responsibility is −0.907, which is significantly negative on the basis of 1%, proving Hypothesis 1; The results of model 3 show that the interaction term between management overconfidence and government environmental governance is 0.259, which is significantly positive on the basis of 5%, proving Hypothesis 2; The results of model 4 show that the interaction item between management overconfidence and media attention is 0.199, which is significantly positive on the basis of 5%, proving Hypothesis 3. The empirical results are consistent with the above regression conclusions, indicating that the research results are robust after selecting random samples.

#### 4.3.2. Replace the Independent Variable Measurement Method

In order to verify the robustness of the regression result, the dummy variable of management overconfidence 0–1, which is an independent variable, is reduced to a continuous variable, and the variables are sorted according to the salary. The management overconfidence is directly measured by the salary proportion of the management. The higher the salary proportion, the stronger the management overconfidence. The test results are shown in Table A2. In model 2, the regression coefficient between management overconfidence and corporate environmental responsibility is −0.522, which is significantly negative on the basis of 1%, proving Hypothesis 1. The results of model 3 show that the interaction term between management overconfidence and government environmental governance is 0.097, which is significantly positive on the basis of 10%, proving Hypothesis 2. The results of model 4 show that the interaction item between management overconfidence and media attention is 0.077, which is significantly positive on the basis of 10%, proving Hypothesis 3. The results are consistent with the above conclusions, indicating that the research results are still robust after the independent variables become continuous variables.

#### 4.3.3. Add Control Variables

Besides the above test methods, considering that in the context of China, the internal management of enterprises may also be burdened with certain environmental protection pressure due to political affiliation and other characteristics [56]. This may have an impact on the environmental responsibility of enterprises. In addition, the institutional investors of an enterprise may also affect the overconfidence of the management of the enterprise due to external supervision, thus affecting the robustness of the empirical results. Therefore, this paper adds two control variables, management’s political connection and institutional investors’ shareholding ratio. Management’s political connection (PC) is expressed as a 0–1 dummy variable, and institutional investors’ shareholding ratio (Ins share) is expressed as the proportion of the number of corporate institutional investors’ shareholding in the total number of corporate shares in the current year. The final test results are shown in Table A3. In model 2, the regression coefficient between management overconfidence and corporate environmental responsibility is −1.125, which is significantly negative on the basis of 1%, proving Hypothesis 1. The results of model 3 show that the interaction term between management overconfidence and government environmental governance is 0.323, which is significantly positive on the basis of 5%, proving Hypothesis 2; The results of model 4 show that the interaction item between management overconfidence and media attention is 0.235, which is significantly positive on the basis of 5%, proving Hypothesis 3. The final results are consistent with the previous research conclusions, indicating that the robustness of the article results is still not affected after adding the two control variables of management political connection and external institutional investors’ shareholding ratio.

#### 4.3.4. Endogeneity

In order to further alleviate the possible endogenous problems of management overconfidence which may interfere with the research conclusions, this paper uses PSM matching method to conduct endogenous test. Firstly, the management overconfidence is grouped according to the median salary. Then, using the proximity matching method and logit regression model, four governance characteristic variables that affect the management’s overconfidence are screened out: board independence (Independence), board Size (BS), management’s number of shareholding (MH), board leadership structure (Duality) and two corporate characteristic variables: corporate assets (SIZE) and property rights (State). Finally, the assumptions are tested step by step based on the matched samples. The final results are shown in Table A4. Model 2 shows that the regression coefficient between management overconfidence and corporate environmental responsibility is −0.963, which is significantly negative on the basis of 1%, proving Hypothesis 1. The results of model 3 show that the interaction term between management overconfidence and government environmental governance is 0.293, which is significantly positive on the basis of 5%, proving Hypothesis 2; The results of model 4 show that the interaction item between management overconfidence and media attention is 0.161, which is significantly positive on the basis of 10%, proving Hypothesis 3. It can be seen that the final assumptions and conclusions have not changed significantly after eliminating the potential pairing bias, indicating that the results of this paper are still robust after considering possible endogenous problems.

### 4.4. Further Analysis: The Impact of Management Overconfidence and Environmental Responsibility on Corporate Value

Previous studies suggested that the management may make irrational decisions due to such overconfidence, such as investing in projects that should not have been invested, even projects with negative net present value [57], carrying out excessive mergers and acquisitions, establishing business empires and other acts [58], causing excessive resource expenditures and waste under the condition of limited enterprise resources, and declining asset returns, thus damaging the interests of the company [3]. Based on the above research, this paper verifies whether this conclusion is also applicable to the management situation of Chinese enterprises. In addition, reference is made to the relevant conclusions on the impact of corporate social responsibility on corporate value in the previous literature, such as ways to increase corporate reputation, attract investors’ attention and reduce financing constraints [59,60,61], this paper further explores whether corporate environmental responsibility in the context of China is also conducive to enhancing corporate value, and whether the negative effect of management overconfidence on corporate value is related to the lack of performance of corporate environmental responsibility.

The test results are reported in Table 6, which studies the effect of management overconfidence and corporate environmental responsibility with logarithm of corporate market value as the proxy variable of corporate value (FV), and model 5 is the same as above. Model 6 is the test of corporate environmental responsibility to corporate value. The regression result shows that the regression coefficient between corporate environmental responsibility and corporate value is 0.040, which is significantly positive on the basis of 1%, indicating that corporate environmental responsibility has a positive impact on corporate value. Model 7 tests the influence of management overconfidence and corporate value. The results show that the regression coefficient between management overconfidence and corporate value is −0.218, and it is significantly negative on the basis of 1%, indicating that management overconfidence will damage corporate value, which is similar to previous research results. Model 8 adds the corporate environmental responsibility to the regression results of management’s overconfidence and corporate value. The results show that after adding the variable of corporate environmental responsibility to the regression, the regression coefficient of management’s overconfidence and corporate value is −0.183, which is still significantly negative on the basis of 1%. Compared with model 7, the absolute value of the regression coefficient is reduced by 0.035, which verifies the partial mediating role of corporate environmental responsibility in the relationship between management’s overconfidence and corporate value and shows that overconfident management’s negative impact on corporate value is partly caused by failure to fulfill corporate environmental responsibility.

## 5. Conclusions

### 5.1. Research Conclusions

Based on the upper echelon theory and stakeholder theory, this paper discusses the influence mechanism and the contingency of management overconfidence on corporate environmental responsibility. The results of the study are as follows. Management overconfidence is related to corporate environmental responsibility. Overconfident management will overestimate the resource endowments of enterprises, tend to challenge decisions with high returns and be interfered with by stakeholders’ opinions, thus forming a psychological exclusion of corporate environmental responsibility. Both government environmental governance and media attention will weaken the negative correlation between management overconfidence and corporate environmental responsibility. In the context of China’s cultural dimension, the government’s environmental governance will bring greater institutional legitimacy pressure to enterprises. In order to obtain more trust and support from the government, the overconfident management will have to change its negative attitude towards corporate environmental responsibility. The increase in media attention has also brought more legitimacy pressure to enterprises. However, different from government environmental governance, media tend to report positively on society, and the public are more likely to attribute the implementation and results of corporate environmental responsibility to corporate managers. The overconfident management has a stronger intention to prove that its ability is higher than average under the attention of the media and will also consider the behavioral consequences of its decision more carefully, will choose to actively strengthen the degree of corporate environmental responsibility corporate environmental responsibility performance. In the further analysis, this paper verifies that the corporate environmental responsibility has a positive impact on corporate value, and the overconfidence of management will damage corporate value, and the corporate environmental responsibility plays a part of intermediary role between the overconfidence of management and corporate value.

### 5.2. Management Implications

First, at the micro level, the impact of management’s overconfidence on the enterprise should be viewed objectively. While encouraging management to give play to its own advantages of optimism and self-confidence, the value loss that management’s irrational decision-making in enterprise practice may bring to the enterprise should also be paid attention to. The board of directors also needs to strengthen the assessment of the psychological characteristics of the management during the selection process of the management. After the management takes office, the board of directors still needs to supervise the management to a certain extent in order to avoid the blind decision made by the management with overconfidence. Second, from the government’s perspective, the government needs to continuously establish and improve the relevant environmental protection laws and regulations system, actively play the role of guidance and supervision, strengthen the corresponding economic means and improve the enterprise environmental protection incentive subsidy policy. Finally, the media should also give full play to the role of information transmitter and regulator and give full play to its professionalism and independence. The media can leverage the management’s awareness of environmental protection from the level of legitimacy by strengthening the long-term continuous supervision of corporate environmental behavior. In addition, the media should promptly report on enterprises with excellent environmental performance to motivate and maintain the management’s determination and motivation to fulfil their environmental responsibilities.

### 5.3. Limitations and Future Research

Firstly, the research chooses Hexun’s environmental responsibility index as the CER data in this paper. Although the index system is relatively comprehensive and scientific, the follow-up research can also construct or adopt a more appropriate environmental responsibility index. Secondly, based on the upper echelon story, the study explores the impact of management overconfidence, a micro-indicator, on CER. We must admit that the research scope needs to be further expanded. Further studies can continue to explore the impact and consequences of other personal characteristics of management based on the upper echelon theory, such as management’s optimism and pessimism, personal experience and other factors and attitudes towards corporate environmental strategies. Finally, this paper divides the environmental legitimacy concerns of external stakeholders into two levels: government environmental governance and media attention. Future research can distinguish the legitimate sources of external stakeholders and explore the differences in corporate decision-making caused by the legitimacy pressure of different stakeholders.

## Figures and Tables

**Figure 1 ijerph-20-00577-f001:**
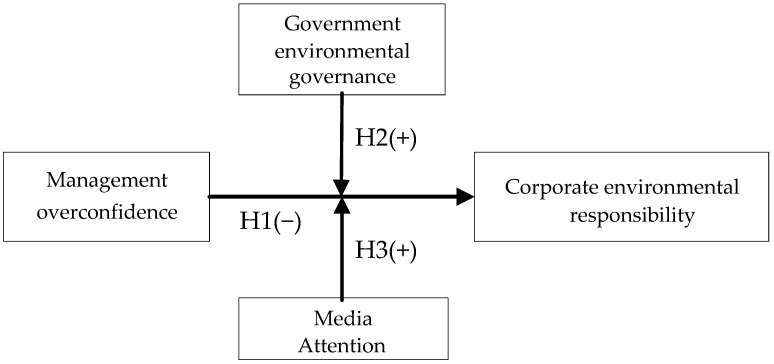
A research framework (developed by authors).

**Table 1 ijerph-20-00577-t001:** Description of variables.

Variables	Definition	Code	Index
Dependent Variable	Corporate environmental responsibility	CER	The environmental responsibility scoring data in the social responsibility assessment system published by Hexun in the current year.
Independent Variable	Management overconfidence	OC	The ratio of the top three management salaries to the sum of all salaries, which is treated as a 0–1 dummy variable based on the median of the ratios.
Moderating Variables	Government environmental governance	EG	Manually collect and sort out the ratio of the frequency of words appearing in the relevant words of the word “environmental protection” in the government work report of each province in China to the full text of the government work report.
	Media attention	Media	Using the advanced search function of Baidu News, the short name of each listed company’s securities is entered one by one and a time-interval search is performed to obtain the total number of corporate news reports for each year, and then one is added to take the natural logarithm.
Control Variables	Management shareholding	MS	The ratio of the number of shares held by management to the total share capital of the Company.
	Board independence	Independence	The ratio of the number of independent directors to the total number of directors of the Company.
	Capital structure	Lev	Year-end gearing ratio of the Company: total liabilities/total assets.
	corporate performance	ROA	Rate of return on assets: net profit after tax/total assets.
	Board leadership structure	Duality	“1” for directors concurrently serving as senior management, or “0” for otherwise.
	Equity concentration	EC	The shareholding ratio of the top ten shareholders.
	Company age	Age	The difference between the year of the current year and the time of establishment of the enterprise.
	Growth of the compony	Growth	The year-end asset growth rate of the enterprise.
	Year	Year	Annual virtual variable.

**Table 2 ijerph-20-00577-t002:** Summary of descriptive statistics.

Variable	Obs	Mean	Std.	P50	Min	Max
CER	7890	2.047	5.511	0.000	0.000	25.000
OC	7890	0.488	0.500	0.000	0.000	1.000
EG	7890	0.015	0.003	0.015	0.006	0.026
Media	7890	2.927	1.289	2.996	0.000	12.660
MS	7890	0.179	0.225	0.035	0.000	0.611
Independence	7890	0.373	0.052	0.333	0.333	0.571
Lev	7890	0.375	0.206	0.352	0.049	1.073
ROA	7890	0.044	0.056	0.043	−0.333	0.216
Duality	7890	0.318	0.466	0.000	0.000	1.000
EC	7890	34.650	14.050	33.250	9.050	73.190
Age	7890	14.700	5.439	15.000	3.000	30.000
Growth	7890	0.267	0.465	0.119	−0.303	2.675

**Table 3 ijerph-20-00577-t003:** The Spearman correlation coefficient between variables.

Variables	1	2	3	4	5	6	7	8	9	10	11	12
1.CER	1.000											
2.OC	−0.086	1.000										
3.EG	−0.110	0.030	1.000									
4.Media	−0.081	−0.024	0.274	1.000								
5.MS	−0.128	−0.008	0.133	0.102	1.000							
6.Independence	0.002	0.052	0.012	0.057	0.044	1.000						
7.Lev	0.128	−0.137	−0.077	0.013	−0.370	−0.001	1.000					
8.ROA	0.028	0.097	0.039	0.119	0.253	−0.019	−0.443	1.000				
9.Duality	−0.091	0.049	0.076	0.047	0.254	0.097	−0.150	0.099	1.000			
10.EC	0.059	0.015	−0.016	−0.029	−0.117	0.052	−0.011	0.107	0.011	1.000		
11.Age	0.001	0.025	0.045	0.231	−0.255	−0.008	0.183	−0.104	−0.106	−0.094	1.000	
12.Growth	−0.054	0.020	0.057	0.156	0.267	−0.002	−0.091	0.366	0.127	0.005	−0.163	1.000

**Table 4 ijerph-20-00577-t004:** The variance inflation factor (VIF) of variables.

Variable	VIF	1/VIF
OC	1.03	0.973
EG	1.07	0.933
Media	1.14	0.876
MS	1.30	0.769
Independence	1.03	0.970
Lev	1.38	0.723
ROA	1.25	0.799
Duality	1.10	0.913
EC	1.04	0.964
Age	1.17	0.852
Growth	1.05	0.950
Mean VIF	1.14	

**Table 5 ijerph-20-00577-t005:** Regression results analysis.

Variables	(1) CER	(2) CER	(3) CER	(4) CER
OC		−0.904 *** (−7.51)	−0.933 *** (−7.68)	−1.323 *** (−4.49)
EG			−0.220 ** (−2.52)	
OC × EG			0.236 ** (2.01)	
Media				0.244 *** (3.36)
OC × Media				0.156 * (1.70)
MS	−0.272 *** (−4.11)	−0.304 *** (−4.59)	−0.304 *** (−4.59)	−0.297 *** (−4.49)
Independence	0.091 (1.52)	0.118 ** (1.98)	0.115 * (1.93)	0.107 * (1.80)
Lev	0.741 *** (10.17)	0.670 *** (9.16)	0.662 *** (9.04)	0.640 *** (8.74)
ROA	0.781 *** (9.17)	0.792 *** (9.33)	0.796 *** (9.38)	0.711 *** (8.24)
Duality	−0.560 *** (−4.21)	−0.522 *** (−3.93)	−0.511 *** (−3.84)	−0.539 *** (−4.06)
EC	0.011 *** (2.61)	0.012 *** (2.68)	0.011 *** (2.64)	0.011 *** (2.61)
Age	0.022 * (1.75)	0.026 ** (2.09)	0.024 ** (1.96)	0.027 ** (2.19)
Growth	−0.337 *** (−3.73)	−0.317 *** (−3.53)	−0.318 *** (−3.53)	−0.344 *** (−3.83)
Year	Control	Control	Control	Control
N	7890	7890	7890	7890
Adj R^2^	0.085	0.092	0.092	0.095
F	50.093	50.821	45.566	46.990

*** is a significant level of 1%. ** is a significant level of 5%. * is a significant level of 10%.

**Table 6 ijerph-20-00577-t006:** Further research.

Variables	(5) CER	(6) FV	(7) FV	(8) FV
OC	−0.904 *** (−7.51)		−0.218 *** (−12.26)	−0.183 *** (−10.64)
CER		0.040 *** (24.72)		0.038 *** (23.91)
MS	−0.304 *** (−4.59)	−0.231 *** (−24.40)	−0.250 *** (−25.60)	−0.238 *** (−25.24)
Independence	0.118 ** (1.98)	0.010 (1.18)	0.020 ** (2.31)	0.016 * (1.86)
Lev	0.670 *** (9.16)	0.424 *** (40.44)	0.436 *** (40.42)	0.410 *** (39.18)
ROA	0.792 *** (9.33)	0.298 *** (24.38)	0.332 *** (26.52)	0.302 *** (24.82)
Duality	−0.522 *** (−3.93)	−0.136 *** (−7.15)	−0.149 *** (−7.62)	−0.129 *** (−6.82)
EC	0.012 *** (2.68)	0.004 *** (7.25)	0.005 *** (7.87)	0.005 *** (7.42)
Age	0.026 ** (2.09)	−0.000 (−0.02)	0.002 (1.00)	0.001 (0.48)
Growth	−0.317 *** (−3.53)	0.025 ** (1.97)	0.017 (1.26)	0.029 ** (2.25)
Year	Control	Control	Control	Control
N	7890	7890	7890	7890
Adj R^2^	0.092	0.446	0.414	0.453
F	50.821	397.229	348.944	385.847

*** is a significant level of 1%. ** is a significant level of 5%. * is a significant level of 10%.

## Data Availability

Not applicable.

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
