# Peer review of "Assessing Influence Mechanism of Management Overconfidence, Corporate Environmental Responsibility and Corporate Value: The Moderating Effect of Government Environmental Governance and Media Attention"

_ijerph, 2022, doi:10.3390/ijerph20010577_

Round 1

Reviewer 1 Report

In this manuscript ,the authors summarize and discuss the current economic and environmental issues facing China's economic development, corporate environmental responsibility, upper echelon theory and stakeholder theory. The author's proposes innovation for this paper is to explore the impact of government and media overconfidence in corporate management on corporate environmental responsibility.

On this basisthis paper presents three sub-argumentsmanagement overconfidence is negatively correlated with corporate environmental responsibilitygovernment environmental governance weakens the negative correlation between management overconfidence and corporate environmental responsibilitymedia attention weakens the negative correlation between management over confidence and corporate environmental responsibility.This paper validates the subarguments and further analyses the the Impact of Management Overconfidence and Environmental Responsibility on Corporate Value.

This work provides new insight and opinion into the development of how to promote better corporate compliance with their environmental responsibilities. one by strengthening the improvement and regulation of the way companies are managed at the top, then the improvement of legal and economic instruments for corporate management by governments, and finally the strengthening of public opinion on companies by the media.

The manuscript is well-orgnized and clearly stated. I would suggesting accepting it after the following concerns are addresed.

1There may be a degree of misrepresentation in the title and the content of the study. The title is Mechanisms of Government Management Influence on Corporate Environmental Responsibility, and the introduction is expressed as exploring the role of government environmental governance and media attention on the moderating effect of managerial overconfidence on the relationship between corporate environmental responsibility. In addition, the relationship between corporate management overconfidence, corporate values and corporate environmental responsibility needs to be further explored in the context of the relationship between government environmental governance and media attention. Does the relationship between corporate management overconfidence, corporate values and corporate environmental responsibility need to be further reflected in the title? It is recommended that the title be further refined to reflect the content of the essay more accurately.

2、“The research selects manufacturing listed companies (industry code: C13-C42) in Shanghai and Shenzhen stock exchanges from 2010 to 2017 as the research sample.The presentation of the data sources is too brief and further refinement of the sources is recommended.the data of government environmental governance and media attention are collected manually by the authorsAre the sources of manually collected government and media-related data reliable? Are the collection methods scientific? Please provide further explanation.

Author Response

Response to Reviewer 1 Comments

The manuscript is well-organized and clearly stated. I would suggesting accepting it after the following concerns are addressed.

1、There may be a degree of misrepresentation in the title and the content of the study. The title is Mechanisms of Government Management Influence on Corporate Environmental Responsibility, and the introduction is expressed as exploring the role of government environmental governance and media attention on the moderating effect of managerial overconfidence on the relationship between corporate environmental responsibility. In addition, the relationship between corporate management overconfidence, corporate values and corporate environmental responsibility needs to be further explored in the context of the relationship between government environmental governance and media attention. Does the relationship between corporate management overconfidence, corporate values and corporate environmental responsibility need to be further reflected in the title? It is recommended that the title be further refined to reflect the content of the essay more accurately.

Response1

Thank you very much for your hard work in the paper review process. Thank you for your impartiality, objectivity and preciseness. Your suggestions are very enlightening to us. According to your advice, as for the title and the content, we have carefully proofread and reviewed the manuscript. The title is changed to Assessing Influence Mechanism of Management Overconfidence, Corporate Environmental Responsibility and Corporate Value——The Moderating Effect of Government Environmental Governance and Media attention.

 2、“The research selects manufacturing listed companies (industry code: C13-C42) in Shanghai and Shenzhen stock exchanges from 2010 to 2017 as the research sample”.The presentation of the data sources is too brief and further refinement of the sources is recommended. “the data of government environmental governance and media attention are collected manually by the authors “Are the sources of manually collected government and media-related data reliable? Are the collection methods scientific? Please provide further explanation.

Response2

According to your suggestion, we added the research sample explanation in the manuscript in 3.1. Data and Sample Selection

3.1. Data and Sample Selection

The research selects manufacturing listed companies (industry code: C13-C42) in Shanghai and Shenzhen stock exchanges from 2010 to 2017 as the research sample, and does the following processing: (1) No significant restructuring phenomenon such as the change of controlling shareholder occurred during the sample observation period; (2) excluding financial sample companies; (3) Excluding ST or *ST companies; (4) eliminating companies with severe data deficiency; (5) To avoid the influence of outliers, the above continuous variables were winsorized. The data of CER are from Hexun CSR report evaluation index, the data of government environmental governance and media attention are collected manually by the authors. All these data are officially released by the enterprise, and their authenticity and reliability are trustworthy. The other data are collected from CSMAR database. In this paper, empirical analysis method is used to verify the hypothesis. With reference to previous research, variables are collected and sorted out, and then a statistical model corresponding to the hypothesis is constructed. Then we choose the STATA16.0 measurement software to perform regression analysis on the selected data. The STATA software has full-featured panel data processing capabilities, and the results show that it is more intuitive, which has been widely used in relevant empirical research.

Thank you very much for your insightful modification suggestions. We feel great thanks for your professional review work on our article.

Best Wishes!

Guiyu Bai

Business School

University of Jinan

Jinan 250002, China

Email: sm_baigy@ujn.edu.cn

Reviewer 2 Report

I commend the authors for undertaking such an innovative study. However, there are several sections in the manuscript that need to be overhauled. 

First, authors should present the Materials and Methods section appropriately and include an in-depth description of the study area (accompanied by a map to ease visualization).

Secondly, authors have merged data analysis with results and discussion which should not be the case. Authors should separate these sections. Data analysis should be taken to the materials and methods section while results and discussion should stand on its own. In addition, results should be adequately interpreted and the discussion of the results should be done in a comparative and analytical fashion making use of related research works. 

Most of the referenced works are not very recent. Authors should source for and reference the most recent works in the domain (2019 to 2022). Authors have referenced so many works dating as far back as the 1990s and early 2000s which does not give an up-to-date picture of study.

Last but not the least, a proper conclusion should be provided as well as the policy implications of the study. 

NB/ Authors should consult the attached file for more details on these comments.

Author Response

Response to Reviewer 2 Comments

Dear Professors,

Thank you very much for your insightful modification suggestions. We feel great thanks for your professional review work on our article. As you are concerned, there are several problems that need to be addressed. According to your nice suggestions, we have made extensive corrections to our previous draft, the detailed corrections are listed below. We hope that the changes we’ve made can resolve all your concerns about the article. If you still have any question about this manuscript, please don’t hesitate to let us know, thank you very much.

1.I commend the authors for undertaking such an innovative study. However, there are several sections in the manuscript that need to be overhauled.

First, authors should present the Materials and Methods section appropriately and include an in-depth description of the study area (accompanied by a map to ease visualization).

Response 1Thank you for your suggestion, in the manuscript we added related content to present the Materials and Methods section appropriately and include an in-depth description of the study area. Please check in the part of Data and Sample Selection

2.Secondly, authors have merged data analysis with results and discussion which should not be the case. Authors should separate these sections. Data analysis should be taken to the materials and methods section while results and discussion should stand on its own. In addition, results should be adequately interpreted, and the discussion of the results should be done in a comparative and analytical fashion making use of related research works.

Response 2

In the manuscript we separate the sections data analysis with results and discussion. In the fourth part we mainly show the results, and we make some changes for adequately interpreted, please check in the manuscript.

  1. Most of the referenced works are not very recent. Authors should source for and reference the most recent works in the domain (2019 to 2022). Authors have referenced so many works dating as far back as the 1990s and early 2000s which does not give an up-to-date picture of study.

Response 3

In the revision version, we have combed the relevant references again, especially the research achievements in recent 3-5 years, especially recent works in the domain (2019 to 2022). We have cited them in the paper and made a lot of modifications in the references. Please refer to them and thank you for your valuable comments

4.Last but not the least, a proper conclusion should be provided as well as the policy implications of the study.

NB/ Authors should consult the attached file for more details on these comments.

Response 4

According to the recommendation of the review experts, based on retaining the original content, we have made a specific scale deletion of the article's research conclusions and management inspiration, from 1048 characters to 666 characters, to better enhance the conciseness of the article. The following are our conclusions and recommendations after deletion:

Based on the upper echelons theory and stakeholder theory, this paper discusses the influence mechanism and the contingency of management overconfidence on corporate environmental responsibility. The results of the study are as follows:

Management overconfidence is related to corporate environmental responsibility. Overconfident management will overestimate the resource endowments of enterprises, tend to challenge decisions with high returns and be interfered with by stakeholders' opinions, thus forming a psychological exclusion of corporate environmental responsibility. Both government environmental governance and media attention will weaken the negative correlation between management overconfidence and corporate environmental responsibility. In the context of China's cultural dimension, the government's environmental governance will bring greater institutional legitimacy pressure to enterprises. In order to obtain more trust and support from the government, the overconfident management will have to change its negative attitude towards corporate environmental responsibility. The increase in media attention has also brought more legitimacy pressure to enterprises. However, different from government environmental governance, media tend to report positively on society, and the public are more likely to attribute the implementation and results of corporate environmental responsibility to corporate managers. The overconfident management has a stronger intention to prove that its ability is higher than average under the attention of the media and will also consider the behavioral consequences of its decision more carefully, will choose to actively strengthen the degree of corporate environmental responsibility corporate environmental responsibility performance.

In the further analysis, this paper verifies that the corporate environmental responsibility has a positive impact on corporate value, and the overconfidence of management will damage corporate value, and the corporate environmental responsibility plays a part of intermediary role between the overconfidence of management and corporate value.

5.2. Management Implications

First, At the micro level, the impact of management's overconfidence on the enterprise should be viewed objectively. While encouraging management to give play to its own advantages of optimism and self-confidence, the value loss that management's irrational decision-making in enterprise practice may bring to the enterprise should also be paid attention to. The board of directors also needs to strengthen the assessment of the psychological characteristics of the management during the selection process of the management. After the management takes office, the board of directors still needs to supervise the management to a certain extent in order to avoid the blind decision made by the management with overconfidence.

Second, from the government's perspective, the government needs to continuously establish and improve the relevant environmental protection laws and regulations system, actively play the role of guidance and supervision, strengthen the corresponding economic means and improve the enterprise environmental protection incentive subsidy policy.

Finally, the media should also give full play to the role of information transmitter and regulator and give full play to its professionalism and independence. The media can leverage the management's awareness of environmental protection from the level of legitimacy by strengthening the long-term continuous supervision of corporate environmental behavior. In addition, the media should promptly report on enterprises with excellent environmental performance to motivate and maintain the management's determination and motivation to fulfil their environmental responsibilities.

Best Wishes!

Guiyu Bai

Business School

University of Jinan

Jinan 250002, China

Email: sm_baigy@ujn.edu.cn

Reviewer 3 Report

The article “Assessing Influence Mechanism of Government Environmental Governance on Corporate Environmental Responsibility” presents a study on the influence of management overconfidence on corporate environmental responsibility. The topic is interesting itself, however, the manuscript contains flaws and needs revision:

1.      First of all, the structure of the manuscript should be revised according to the journal rules. Please follow the Instruction for authors https://www.mdpi.com/journal/ijerph/instructions. Sections Materials & Methods, Results, Conclusions are mandatory.

2.      In Abstract please formulate the aim of the research.

3.      The article is not well-presented and is not easy to follow. Please improve the quality of Figure and Tables.

4.      Please reformat page 11, excluding the empty space.

5.      Some of the tables could be placed in Appendix.

6.      Please indicate clear the research novelty of the paper.

7.      The major concern is regarding the main conclusion. The authors’ statement that “Management overconfidence has a negative correlation with corporate environmental responsibility” looks questionable. The matter depends on the level of management professionalism. The intellectual management, committed to environmental ideas, can be overconfident, and this will not have the negative impact.

Author Response

Response to Reviewer 3 Comments

Dear Professor:

Many thanks for your comments about our manuscript ijerph-2080905. Thank you very much for your affirmation and guidance. Overall the comments have been fair, encouraging and constructive. We have learned much from it. After carefully studying the reviewer’ comments and your advice, we have made corresponding changes to the paper. Our response of the comments is as follows:

Point 1:  First of all, the structure of the manuscript should be revised according to the journal rules. Please follow the Instruction for authors https://www.mdpi.com/journal/ijerph/instructions. Sections Materials & Methods, Results, Conclusions are mandatory.

Response 1: Thank you very much for your valuable suggestions. We carefully follow the Instruction for authors https://www.mdpi.com/journal/ijerph/instructions. and make changes for the Sections of our manuscript,the new section title include:Materials & Methods, Results, Conclusions.

Point 2: In Abstract please formulate the aim of the research.

Response 2: Thank you very much for your advice. We formulate the aim of the research and add it into the abstract. The new abstract is as follows:

China's economic development has gradually entered a new period of slowing down and changing from quantity to quality, which has put forward higher requirements for environmental quality. How to better fulfill environmental responsibilities and realize a virtuous circle of "environmental protection for development" and a value growth model are essential issues that enterprises should consider and solve. In order to provide more empirical evidence on internal governance for the influencing factors of corporate environmental responsibilities decisions and promote enterprise to make more effective environmental responsibility decision-making. Based on upper echelon theory and stakeholder theory, focusing on the micro-situation of the corporate, this paper empirically tests the influence of management overconfidence on corporate environmental responsibility by using the OLS regression analysis method, taking the manufacturing listed companies in Shanghai and Shenzhen Stock Exchange of China from 2010 to 2017 as the research sample, and discusses the moderating effect of government environmental governance and media attention on the relationship between management overconfidence and corporate environmental responsibility . The empirical results show a negative correlation between management overconfidence and corporate environmental responsibility. Both government environmental governance and media attention will weaken the negative correlation between management overconfidence and corporate environmental responsibility. Further research finds that management overconfidence has a weakening effect on corporate value, and corporate environmental responsibility plays a partial mediating role between management overconfidence and corporate value.

Point 3: The article is not well-presented and is not easy to follow. Please improve the quality of Figure and Tables.

Response 3: Your suggestion really means a lot to us. Yes, it would be more understandable if we re-optimize the Figure and Tables in this paper. We redraw the Figure, modify the previous hypothesis order, and simplify and optimize the Tables in this paper.

Point 4: Please reformat page 11, excluding the empty space.

Response 4: Thanks for your suggestion. Page 11 has been reformatted, excluding the empty space, and we have proofread the format of the text. Thank you again for pointing out this error.

Point 5: Some of the tables could be placed in Appendix.

Response 5: Thank you for your suggestion. We have moved the tables in the robustness test section (Table6-Table9) to Appendix A to improve the readability of the article.

Point 6: Please indicate clear the research novelty of the paper.

Response 6: Thank you for pointing out this point. In this paper, we combine the research novelty with the research contribution to improve the conciseness of the introduction. To better respond to your suggestions, we re-discussed this paper's research innovation and contribution, emphasizing and highlighting the research novelty. The 2 and 3 pages of the article are our revised contents:

Compared with the existing literature, the possible research contributions and research innovations are as follows: Firstly, the research explores the psychological characteristics of the management—overconfidence's attitude towards the enterprise's environmental decision-making, which not only can expand the application paradigm of the upper echelon theory and enrich the management decision-making research literature in the field of management's personality characteristics, but also makes some exploration and innovation on how enterprises view the current environmental problems from the corporate governance field and deepens the understanding of the influencing factors of CER at the micro level. Secondly, based on the stakeholder theory, we consider the legitimacy problems faced by enterprises. Given the institutional situation in China, we try to distinguish the external stakeholders in variable selection and measurement from the government's perspective and external media attention. By exploring the requirements of different stakeholders for continuous improvement of the environment and the differential impact mechanism of external legitimacy pressure on corporate strategic decisions, we clarify the role path of external stakeholders in corporate governance and deepen and innovate the logic of current literature research on the impact of external stakeholders on corporate decision-making. Finally, our research conclusion not only expands the upper echelon theory and stakeholder theory on the theoretical level but also provides new ideas and empirical evidence for developing countries with similar institutional and cultural patterns on how to view the relationship between executive personality and corporate decision-making.

Point 7: The major concern is regarding the main conclusion. The authors’ statement that “Management overconfidence has a negative correlation with corporate environmental responsibility” looks questionable. The matter depends on the level of management professionalism. The intellectual management, committed to environmental ideas, can be overconfident, and this will not have the negative impact.

Response 7: Thank you for this suggestion. It would have been interesting to explore this aspect.

We quite agree with you that management committed to environmental protection will not reduce their willingness to fulfil their environmental responsibilities if they tend to be overconfident. However, our research conclusions are mainly based on the following considerations: 1. We mainly explore the psychological state of overconfidence of the management. A large number of studies have proved that the psychological state of overconfidence will overestimate the internal resources of the enterprise and its management ability, thus ignoring the legitimacy pressure of external stakeholders, which provides a novel perspective for our research. We draw lessons from the more mature theories in management, connect the external environmental responsibility with the environmental protection demands of stakeholders, and conclude that overconfident management will have a negative attitude towards corporate environmental responsibility through rigorous theoretical and logical discussion. 2. In our study, we selected a large number of listed company data as samples, and through a series of rigorous statistical regression methods, we verified that management overconfidence and corporate environmental responsibility have a negative correlation in the sample as a whole. Judging from the regression results, the 7890 groups of samples we selected have accepted the logical arguments we put forward in this paper and also conform to the normative statistical analysis.

In response to your suggestion, we cannot deny that: 1. The results we have obtained cannot fully demonstrate the negative correlation between management overconfidence and corporate environmental responsibility in all samples. Your suspicion is reasonable to some extent, but our conclusions do not violate the general assumptions of statistics and enterprise resource-based views. 2. Your proposal has provided us with novel and interesting research thinking for our research. We will consider the management ability and environmental protection concept of management in our subsequent research and explore how the management ability and psychological characteristics jointly affect the business decision-making of the enterprise. Thank you again for your valuable advice.

Thank you very much for your affirmation and guidance. This is our revision instructions. We have also made specific revisions in the paper. Please review them. If you have any questions, please point out that we will accept and revise them with open mind, Wish you all the best.

Sincerely yours

Guiyu Bai

Business School

University of Jinan

Jinan 250002, China

Email: sm_baigy@ujn.edu.cn

Round 2

Reviewer 3 Report

I appreciate the revision done to improve the manuscrit, however, some minor changes still are required.

1. The text added at lines 15-17 is unclear and needs editing.

2. Please exclude the bold letters at lines 336-348

3. Conclusions should be re-written as a single section.

4.The manuscript should be properly written in a third person (e.g.limes 101,109, etc.needs revision).

Author Response

Response to Reviewer 3 Comments

Dear Professor:

We feel great thanks for your professional review work on our article. As you are concerned, there are several problems that need to be addressed. According to your nice suggestions, we have made extensive corrections to our previous draft, the detailed corrections are listed below:

I appreciate the revision done to improve the manuscript, however, some minor changes still are required.

Point 1: The text added at lines 15-17 is unclear and needs editing.

Response 1: Thank you for pointing this out. In response to your suggestion, we rewrote lines 15-17 to better highlight the research theme and purpose of the article. The following is the revised content:

China's economic development has gradually entered a new period of slowing down and changing from quantity to quality, which has put forward higher requirements for environmental quality. How to better fulfill environmental responsibilities and realize a virtuous circle of "environmental protection for development" and a value growth model are essential issues that enterprises should consider and solve. Overconfidence, as one of the significant psychological characteristics of management, has caused more and more attention to its economic consequences. In order to clarify the internal logical relationship between the two and help enterprises optimize their environmental responsibility decisions, the paper is based on upper echelon theory and stakeholder theory, focusing on the micro-situation of the corporate,  empirically testing the influence of management overconfidence on corporate environmental responsibility by using the OLS regression analysis method, taking the manufacturing listed companies in Shanghai and Shenzhen Stock Exchange of China from 2010 to 2017 as the research sample, and discusses the moderating effect of government environmental governance and media attention on the relationship between management overconfidence and corporate environmental responsibility. The empirical results show a negative correlation between management overconfidence and corporate environmental responsibility. Both government environmental governance and media attention will weaken the negative correlation between management overconfidence and corporate environmental responsibility. Further research finds that management overconfidence has a weakening effect on corporate value, and corporate environmental responsibility plays a partial mediating role between management overconfidence and corporate value.

Point 2: Please exclude the bold letters at lines 336-348

Response 2:

According to the suggestion, we excluded inappropriate bold letters in the text and carefully checked the full paper.

Point 3: Conclusions should be re-written as a single section.

Response 3: As suggested by the reviewer and referring to the research of other similar topics in this journal, we have integrated the paragraphs in Conclusion, and integrated the Research Conclusions and Management Implications paragraphs into a single paragraph to optimize the overall structure of the article.

Point 4: The manuscript should be properly written in a third person (e.g.lines 101,109, etc.needs revision).

Response 4: We agree with the reviewer's assessment. Accordingly, throughout the manuscript, we have revised the inappropriate person in the text to the third person. Some of our modified statements read as follows:

Compared with the existing literature, the possible research contributions and research innovations are as follows: Firstly, the research explores the psychological characteristics of the management—overconfidence's attitude towards the enterprise's environmental decision-making, which not only can expand the application paradigm of the upper echelon theory and enrich the management decision-making research literature in the field of management's personality characteristics, but also explores the motivation of corporate environmental responsibility decision-making in corporate governance and deepens the understanding of the influencing factors of CER at the micro level.

By exploring the requirements of different stakeholders for continuous improvement of the environment and the differential impact mechanism of external legitimacy pressure on corporate strategic decisions, the research clarifies the role path of external stakeholders in corporate governance and deepen and innovate the logic of current literature research on the impact of external stakeholders on corporate decision-making. Finally, the research conclusion not only expands the upper echelon theory and stakeholder theory on the theoretical level but also provides new ideas and empirical evidence for developing countries with similar institutional and cultural patterns on how to view the relationship between executive personality and corporate decision-making.
